# Autoantibody Release in Children after Corona Virus mRNA Vaccination: A Risk Factor of Multisystem Inflammatory Syndrome?

**DOI:** 10.3390/vaccines9111353

**Published:** 2021-11-18

**Authors:** Reiner Buchhorn, Carlotta Meyer, Kai Schulze-Forster, Juliane Junker, Harald Heidecke

**Affiliations:** 1Department of Pediatrics, Caritas-Krankenhaus Bad Mergentheim, Uhlandstraße 7, 97980 Bad Mergentheim, Germany; 2Praxis für Kinder- und Jugendmedizin, Kinderkardiologie und Erwachsene Mit Angeborenen Herzfehlern, Am Bahnhof 1, 74670 Forchtenberg, Germany; 3CellTrend GmbH, 14943 Luckenwalde, Germany; Meyer@CellTrend.de (C.M.); schufo@celltrend.de (K.S.-F.); junker@celltrend.de (J.J.); heidecke@celltrend.de (H.H.)

**Keywords:** COVID-19, autoantibodies, children, multisystem inflammatory syndrome, SARS-CoV-2 vaccination, Hashimoto thyroiditis

## Abstract

Multisystem inflammatory syndrome (MIS) is a new systemic inflammatory acute onset disease that mainly affects children (MIS-C) and, at a lesser frequency, adults (MIS-A); it typically occurs 3–6 weeks after acute SARS-CoV infection. It has been postulated and shown in adults that MIS may occur after SARS-CoV-2 vaccination (MIS-V). Our current case is one of the first published cases with a multisystem inflammatory syndrome in an 18-year-old adolescent after the SARS-CoV-2 vaccine from Pfizer/BionTech (BNT162b2), who fulfills the published level 1 criteria for a definitive disease: age < 21 years, fever > 3 consecutive days, pericardial effusion, elevated CRP/NT-BNP/Troponin T/D-dimeres, cardiac involvement, and positive SARS-CoV-2 antibodies. The disease starts 10 weeks after the second vaccination, with a fever (up to 40 °C) and was treated with amoxicillin for suspected pneumonia. The SARS CoV-2-PCR and several antigen tests were negative. With an ongoing fever, he was hospitalized 14 days later. A pericardial effusion (10 mm) was diagnosed by echocardiography. The C-reactive protein (174 mg/L), NT-BNP (280 pg/mL), and Troponin T (28 pg/mL) values were elevated. Due to highly elevated D-dimeres (>35,000 μg/L), a pulmonary embolism was excluded by thoracal computer tomography. If the boy did not improve with intravenous antibiotics, he was treated with intravenous immunoglobulins; however, the therapy was discontinued after 230 mg/kg if he developed high fever and hypotension. A further specialized clinic treated him with colchicine and ibuprofen. The MIS-V was discovered late, 4 months after the onset of the disease. As recently shown in four children with MIS-C after SARS-CoV-2 infection and a girl with Hashimoto thyroiditis after BNT162b2 vaccination, we found elevated functional autoantibodies against G-protein-coupled receptors that may be important for pathophysiology but are not conclusive for the diagnosis of MIS-C. Conclusion: We are aware that a misattribution of MIS-V as a severe complication of coronavirus vaccination can lead to increased vaccine hesitancy and blunt the global COVID-19 vaccination drive. However, the pediatric population is at a higher risk for MIS-C and a very low risk for COVID-19 mortality. The publication of such cases is very important to make doctors aware of this complication of the vaccination, so that therapy with intravenous immunoglobulins can be initiated at an early stage.

## 1. Introduction

Multisystem inflammatory syndrome (MIS) is a new systemic inflammatory acute onset disease that mainly affects children (MIS-C) and, at a lesser frequency, adults. The disease usually occurs in children 3–6 weeks after acute SARS-CoV infection. In Februrary of 2021, case definition & guidelines for data collection, analysis, and presentation of immunization safety data from a Brighton Collaboration was published in the Journal *Vaccine* [1]. At this time, the coronavirus vaccine was mainly used in adults and no case of multisystem inflammatory syndrome in children (MIS-C) after vaccination was published. However, in July 2021 the first three cases with multisystemic inflammatory syndrome in adults (MIS-A) after SARS-CoV-2 vaccination were published [2]. The vaccine was not specified in this publication, but two of these patients had COVID-19 disease, shortly before vaccination (34 days and 43 days), and the disease started with a short delay 4 days after the second vaccine or 19 days after the first vaccine. All patients survived and were treated with Methylprednisolone and antibiotics. Another case of MIS-A after vaccination was published in July 2021, and the authors proposed the term multisystem inflammatory syndrome after vaccination (MIS-V) [3].

On 2 August 2021, the Danish Medicines Agency investigated a case of inflammatory conditions, reported after COVID-19 vaccination, in a 17-year-old boy, after receiving COVID-19 vaccine from Pfizer/BionTech (BNT162b2) (Mainz, Germany) (https://www.ema.europa.eu/en/news/meeting-highlights-pharmacovigilance-risk-assessment-committee-prac-30-august-2-september-2021, accessed on 3 September 2021, published by European Medicines Agency, Domenico Scarlattilaan 6, 1083 HS Amsterdam, The Netherlands).

Our first case report demonstrates a severe inflammatory disease in an 18-year-old boy with high fever, as well as pericardial and pleural effusions, ten weeks after the second COVID-19 vaccine from Pfizer/BionTech (BNT162b2), who fulfills the published MIS-C Level 1 Criteria of Diagnostic Certainty [1].

Myocarditis became a hallmark for complications not only in COVID-19 but also as an unwanted side effect of the coronavirus mRNA vaccination [3]. A recent review summarizes and evaluates the available evidence on the pathogenesis, diagnosis, and treatment of myocarditis and inflammatory cardiomyopathy, with a special focus on virus induced myocarditis [4]. Beta adrenoreceptor autoantibodies seem to be important for pathophysiology with therapeutic implications [5]. We measured elevated autoantibodies against G-protein-coupled receptors in children with multisystem inflammatory syndrome (MIS-C) after a natural SARS-CoV-2 infection [6]. The data are in accordance with multiple elevated autoantibodies after SARS-CoV-2 infections in adults [7,8]. We now publish these autoantibodies in an 18-year-old boy with severe inflammatory disease after coronavirus mRNA vaccination and prove the release of these autoantibodies against G-protein-coupled receptors in a girl with Hashimoto thyroiditis after coronavirus mRNA vaccination (Pfizer-BioNTech BNT162b2).

The anti-adrenergic receptors (α1, α2, β1, β2), anti-muscarinic receptors (M1- M5), anti-endothelin receptor type A, and anti-angiotensin II type 1 receptor autoantibodies were measured in serum samples using a sandwich ELISA kit (CellTrend GmbH Luckenwalde, Germany). The microtiter 96-well polystyrene plates were coated with G-protein-coupled receptors. To maintain the conformational epitopes of the receptor, 1 mM calcium chloride was added to every buffer. Duplicate samples of a 1:100 serum dilution were incubated at 4 °C for 2 h. After washing steps, plates were incubated for 60 min with a 1:20,000 dilution of horseradish peroxidase-labeled goat anti-human IgG, used for detection. In order to obtain a standard curve, plates were incubated with test serum from an anti-G-protein-coupled receptors autoantibody positive index patient. The ELISAs were validated, according to the national standards (DIN EN ISO 138485:2016) and FDA’s “Guidance for industry: Bioanalytical method validation”.

## 2. Case Report 1

The 18-year-old boy suffers from hypoxic ischemic encephalopathy after a complicated birth and receives pharmacotherapy, due to his epilepsy (clobazam, oxcarbazein, and rufinamid) and tetraspastic (baclofen). Since he is classified as a high-risk patient for COVID-19, he was vaccinated (BNT162b2) for the first time shortly after the vaccine was approved in January 2021. He had no relevant side effects and got his second vaccination in February 2021. Ten weeks after this vaccination, he developed a high fever (up to 40 °C) and was treated with amoxicillin for a suspected pneumonia. SARS CoV-2-PCR and several antigen tests were negative. With ongoing fever, he was hospitalized 14 days later, the SARS CoV-2-PCR was negative, again, at admission. A pericardial effusion (10 mm) was diagnosed by echocardiography and computer tomography. The C-reactive protein was highly elevated (174 mg/L), the NT-BNP (280 pg/mL) and Troponin T (28 pg/mL) values are elevated. Due to highly elevated D-dimeres (>35,000 μg/L), the pulmonary embolism was excluded by thoracal computer tomography. As the boy did not improve with intravenous antibiotics, he was treated with intravenous immunoglobulins, but the therapy was stopped after 230 mg/kg, as he developed a high fever and hypotension. The patient was then transferred to a university clinic, which initiated therapy with colchicine and ibuprofen, during which the symptoms slowly improved. The pericardial effusion disappeared, and he was presented to our practice for a follow-up appointment, with the question of whether a third vaccination could be administered. We found no effusions in echocardiography, the C-reactive protein and Troponin T values were not elevated, and the SARS-CoV-2 antibodies were positive.

Due to the long delay (10 weeks) between the vaccination and the multisystem inflammatory syndrome (MIS-C), the diagnosis was made late and remained unsure. However, the guidelines defined a time frame for onset of MIS-C lower than 12 weeks post-infection/vaccination, although MIS-C cases predominantly present 4–6 weeks following COVID-19. Moreover, a specialized clinic did not find any other explanation for the inflammatory syndrome of this boy, which was not doubted. It clearly corresponds to our duty of care, that we reported the unwanted side effect of BNT162b2 to the responsible drug authorities.

There is a high burden of inflammatory disease in the family: the brother of the patient had pericardial effusion, of unknown origin, six years ago, which was treated with colchicine for 6 months. He received a coronavirus vaccination (BNT162b2), without significant side effects or relapse of his pericardial effusion. The mother of the boy recently suffered from inflammation with pericardial effusions without COVID-19 or vaccination. Some genetic autoinflammatory syndrome was excluded by the university hospital (tumor necrosis factor receptor associated periodic syndrome, familial mediterranean fever).

Based upon our report, regarding elevated functional autoantibodies against G-protein-coupled receptors in children with MIS-C, we performed this autoantibody analysis in a research laboratory and found a comparable pattern of autoantibodies in the current patient after SARS-CoV-2 vaccination (Table 1, Figure 1). In detail, anti-angiotensin 1 receptor, anti-endothelin receptor, anti-α1 adrenergic receptor, anti-β1 adrenergic receptor, anti-β2 adrenergic receptor, and anti-muscarinic cholinergic receptor-2/3/4 autoantibodies were significantly elevated after the SARS-CoV-2 vaccination.

## 3. Case Report 2

The thirteen-year-old girl had a pacemaker implantation at the age of one year, due to recurrent syncopes, due to sinus arrest, with asystole up to 10 s. With atrial pacing (AAI mode), the syncopes disappeared, but later she developed intermittent atrioventricular block; the pacemaker was updated to a dual chamber pacing (DDD mode) at the age of 8 years. At the age of 13, she had elevated thyroid stimulating hormone (TSH) in blood screening and Hashimoto thyroiditis was diagnosed (by highly elevated thyroid peroxidase antibodies (anti-TPO) and an ultrasound scan), treated with thyroxin, and selen supplementation. We anticipated an immunological cause for her arrhythmia, as four of the autoantibodies against G-protein-coupled receptors were significantly elevated (Figure 2) [9].

Together with her parents, she decided on a coronavirus mRNA vaccination (Pfizer-BioNTech BNT162b2); however, because of her autoimmune disease, the release of autoantibodies should be measured. After the first coronavirus mRNA vaccination, autoantibodies against G-protein-coupled receptors further uniformly increased, and antithyroid peroxidase antibodies (Anti-TPO) slightly increased (Table 1, Figure 2). In detail, anti-angiotensin 1 receptor, anti-endothelin receptor, anti-α1 adrenergic receptor, anti-β1 adrenergic receptor, anti-β2 adrenergic receptor, and anti-muscarinic cholinergic receptor-3/4 autoantibodies were significantly elevated after SARS-CoV-2 vaccination. If the girl had no clinical side effects of the vaccination, she would get her second vaccination four weeks later. Six weeks after the second vaccination, anti-TPO further increased, while autoantibodies against G-protein-coupled receptors returned to values slightly above the baseline values. In this time, pacemaker monitoring showed an increase of atrial pacing from 8.7% to 10.5% after the first vaccination and to 19.4% after the second vaccination. Ventricular pacing increased from <0.1% to 0.2% after vaccination, which may indicate an atrioventricular blockade. She had normal T3 and T4 values, but the thyroid stimulating hormone (TSH) increased from 2.99 μU/mL to 8.65 μU/mL after the second vaccination, and the thyroid hormone supplementation must be elevated.

## 4. Discussion

After the Danish case (https://www.ema.europa.eu/en/news/meeting-highlights-pharmacovigilance-risk-assessment-committee-prac-30-august-2-september-2021; accessed on 3 September 2021), our first case is the second published case with a multisystem inflammatory syndrome (MIS-C) in an adolescent after SARS-CoV-2 vaccination, who fulfills the published level 1 criteria for a definitive disease: age < 21 years, fever > 3 consecutive days, pericardial and pleural effusions, weakness, elevated CRP/NT-BNP/Troponin T/D-dimere, cardiac involvement, SARS-CoV-2 antibodies, time frame < 12 weeks.

As recently shown in children with immunological complications after SARS-CoV-2 infections [6], he had a comparable pattern of increased autoantibodies against G-protein-coupled receptors (Table 1, Figure 1). Most of all, anti-β adrenergic receptor autoantibodies have been shown to be elevated in adults with heart failure [5] and children with cardiomyopathies [10]. However, anti-α1 and anti-muscarinergic receptor 3 + 4 autoantobodies were more elevated in four children after SARS-CoV-2 infection, compared to this case. The number of cases was too low to be able to draw general conclusions from this. Moreover, we do not know the baseline values (before vaccination) in the boy who had pre-existing illness.

To prove our hypothesis of BNT162b2 vaccination-induced autoantibody release in children, we measured the autoantibody against G-protein-coupled receptors in a girl with Hashimoto thyroiditis after vaccination (Case 2, Figure 2). We found a uniform increase of all these autoantibodies after the first vaccination, which returned to baseline six weeks after the second vaccination, but thyroid peroxidase autoantibodies further increased. She had no clinical side effects, but pacemaker monitoring showed an impairment of her arrhythmia, known of since early childhood and currently successfully treated with the dual chamber pacemaker. Moreover, while TPO antibodies significantly increased after vaccination, we had to increase her thyroxin treatment to normalize the elevated TSH values.

With these cases, we try to connect knowledge about the potential of SARS-CoV-2 to trigger autoimmunity [7] with known cardiovascular complications of the disease and vaccination. Autoinflammation may explain the impact of SARS-CoV-2 infections on Hashimoto thyroiditis [11], as well as arrhythmogenesis [9] and myocarditis [10].

At least, it seems not to be the whole virus but the spike protein that induces autoimmunity, the most imminent danger for children in this pandemic. Together with our published data about elevated functional autoantibodies against G-protein-coupled receptors in children with MIS-C, the current case reports after vaccination show a comparable effect on the network of autoantibodies, and different autoantibodies uniformly shift to enhanced blood levels after the immunological response to the vaccine, comparable to our data after the natural infection [6]. In both cases, the spike protein is the target of the immune response. We tried to explain the effect of the spike protein on multiple autoantibody pathways, with a blockade of the so called cholinergic anti-inflammatory pathway [12]. As recently published, COVID-19-related myocarditis may be related to the cholinergic anti-inflammatory pathway [13]. The nicotinergic acetyl-choline receptor alpha7 subunit (α7nAChR) is cardinal in this pathway and is mainly expressed on the membrane of immune cells [14]. There is some evidence of a molecular mimicry of nicotinergic acetyl-choline receptor alpha7 subunit and the spike protein [15], which could explain the proinflammatory effect by a blockade of the anti-inflammatory pathway that is neuronally controlled by the vagus nerve.

We investigated the impact of the autonomic nervous system on SARS-CoV-2 infection by using an analysis of heart rate variability and found very low heart rate variabilities during the acute disease and the multisystemic inflammatory syndrome (MIS-C) [16,17], which indicates low vagus activity. A low heart rate variability in the elderly and patients with chronic diseases may explain the higher risk of death from SARS-CoV-2 infections. We have developed an algorithm, based on a 24 h analysis of heart rate variability, to estimate the risk of death from COVID-19 [18].

The publication of the current cases is very important, in order to make doctors aware vaccination complications, such as MIS-C, if therapy with intravenous immunoglobulins can be initiated at an early stage. This awareness is essential when vaccinating children and adolescents who are not at increased risk of death from COVID-19. When informing the parents, we should refrain from claiming that the vaccination protects against MIS-C, as shown in the first case. Laboratory tests of autoantibodies, against G-protein-coupled receptors, may help to understand pathophysiology, but these tests are not conclusive for diagnosis of MIS-C, if the girl with Hashimoto disease had a comparable increase of the antibodies but no clinical signs of MIS-C. We would like to emphasize, again, that the sample size is too small, as well as lacking in proper methodology (levels of autoantibodies before vaccination in case 1) to arrive at a final conclusion. However, our data has encouraged us to prospectively investigate the formation of autoantibodies against G-protein-coupled receptors after various SARS-CoV-2 vaccinations.

Many countries are in the process of approving the vaccination for all children because the health policy goal of achieving herd immunity seems to only be achieved by immunizing the population as completely as possible. Although the complications of the vaccination are apparently rare, we may focus on the importance of management and therapy of these rare cases if we want to maintain public acceptance of the SARS-CoV-2 vaccination. We are aware that a misattribution of MIS-C as a severe complication of coronavirus vaccination can lead to increased vaccine hesitancy and blunt the global COVID-19 vaccination drive. However, the pediatric population is at a higher risk for MIS-C and a very low risk for COVID-19 mortality. At the currently high infection rate, the vaccination decision in childhood should be made dependent on the risk assessment of autoinflammatory diseases of COVID-19, compared with the vaccination. Recent observational data from Clalit Health Services, the largest health care organization in Israel, shows a lower risk of myocarditis after BNT162b2mRNA vaccination (risk ratio, 3.24; confidence interval, 1.55 to 12.44; risk difference, 2.7 events per 100,000 persons), compared to COVID-19 (18.28; 95% CI, 3.95 to 25—12; risk difference [11], 0 events per 100,000 persons) [3].

## 5. Conclusions

Multisystemic inflammatory syndrome seems to be a complication after COVID-19 and probably to a lesser frequency after SARS-CoV-2 vaccination. These complications after COVID-19 and SARS-CoV-2 vaccination may be related to autoimmunity. However, elevated G-Protein-coupled autoantibodies as in our cases are not clearly related to clinical symptoms and must be prospectively proofed after vaccination.

## Figures and Tables

**Figure 1 vaccines-09-01353-f001:**
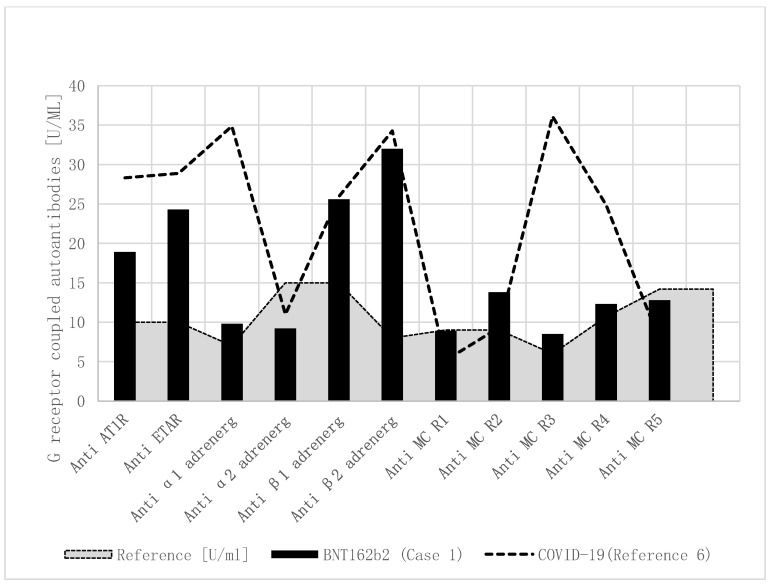
Functional autoantibodies against G-protein-coupled receptors in children with MIS-C after COVID-19 and BNT162b2. Functional autoantibodies against G-protein-coupled receptors in a boy with MIS-C after BNT162b2 vaccination (Case1), and the mean of four published children with MIS-C after COVID-19. Compared to healthy controls, anti-angiotensin 1 receptor, anti-endothelin, and anti-β adrenergic receptor autoantibodies were elevated. Additionally, elevated anti-α1 adrenergic receptor and anti-muscarinic cholinergic receptor-3/4 autoantibodies were only measured in the four children with MIS-C after COVID-19. **Anti-AT1R**: anti-angiotensin 1 receptor autoantibody; **anti-ETAR**: anti-endothelin receptor autoantibody; **anti-α1 adrenerg**: anti-α1 adrenergic receptor autoantibody; **anti-α2 adrenerg**: anti-α2 adrenergic receptor autoantibody; **anti-β1 adrenerg**: anti-β1 adrenergic receptor autoantibody; **anti-β2 adrenerg**: anti-β2 adrenergic receptor autoantibody; **anti-MC R1**: anti-muscarinic cholinergic receptor-1 autoantibody; **anti-MC R2**: anti-muscarinic cholinergic receptor-2 autoantibody; **anti-MC R3**: anti-muscarinic cholinergic receptor-3 autoantibody; **anti-MC R4**: anti-muscarinic cholinergic receptor-4 autoantibody; **anti-MC R5**: anti-muscarinic cholinergic receptor-5 autoantibody.

**Figure 2 vaccines-09-01353-f002:**
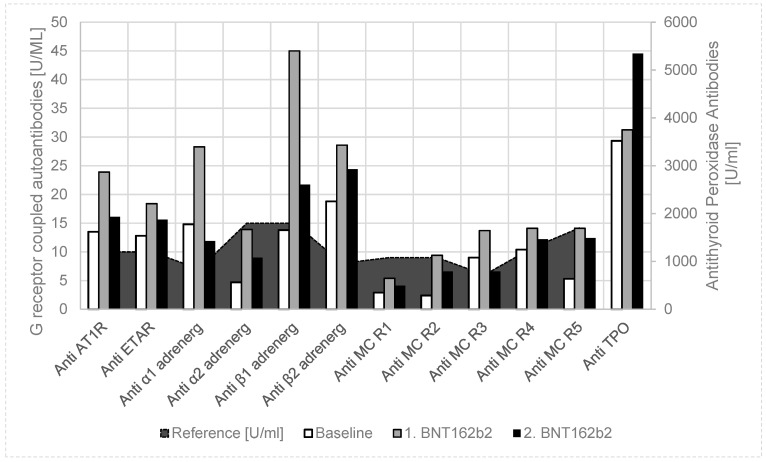
Case 2: Autoantibody release after Pfizer-BioNTech BNT162b2 vaccination in a 13-year-old girl with Hashimoto thyroiditis. Compared to healthy controls and the baseline values, we found a uniform increase of functional autoantibodies against G-protein-coupled receptors and antithyroid peroxidase antibodies in a girl with a known autoimmune disease (after the first BNT162b2 vaccination); however, there was a further increase of only the antithyroid peroxidase antibodies after the second vaccination. **Anti-AT1R**: anti-angiotensin 1 receptor autoantibody; **anti-ETAR**: anti-endothelin receptor autoantibody; **anti-α1 adrenerg**: anti-α1 adrenergic receptor autoantibody; **anti-α2 adrenerg**: anti-α2 adrenergic receptor autoantibody; **anti-β1 adrenerg**: anti-β1 adrenergic receptor autoantibody; **anti-β2 adrenerg**: anti-β2 adrenergic receptor autoantibody; **anti-MC R1**: anti-muscarinic cholinergic receptor-1 autoantibody; **anti-MC R2**: anti-muscarinic cholinergic receptor-2 autoantibody; **anti-MC R3**: anti-muscarinic cholinergic receptor-3 autoantibody; **anti-MC R4**: anti-muscarinic cholinergic receptor-4 autoantibody; **anti-MC R5**: anti-muscarinic cholinergic receptor-5 autoantibody.

**Table 1 vaccines-09-01353-t001:** Autoantibodies after vaccination, compared to children with MIS-C after COVID-19, as recently published [6].

	Anti-AT1R	Anti-ETAR	Anti-α1 Adrenerg	Anti-α2 Adrenerg	Anti-β1 Adrenerg	Anti-β2 Adrenerg	Anti-MC R1	Anti-MC R2	Anti-MC R3	Anti-MC R4	Anti-MC R5	TPO
**Healthy Control** **95% Perc. [U/mL]**	<10	<10	<7	<15	<15	<8	<9	<9	<6	<10.7	<14.2	<60
**At risk 90% Perc.**	10–17	10–17	7–11			8–14			6–10			
**MIS-C after COVID-19 (N = 4)**												
**Mean from Reference 6**	28.3	28.8	34.8	11	25.9	34.2	5.1	9.3	36.1	24.8	7.7	
**MIS-C after BNT162b2 vaccination (Case 1)**												
**Case 1**	18.9	24.3	9.8	9.2	25.6	32.0	8.9	13.8	8.5	12.3	12.8	
**Hashimoto Thyreoditis (Case 2)**												
**Baseline**	13.5	12.8	14.8	4.7	13.8	18.8	2.9	2.4	9.0	10.4	5.3	3522
**1. Biontec**	23.9	18.4	28.3	13.9	45.0	28.6	5.4	9.4	13.7	14.1	14.1	3751
**2. Biontec**	16.1	15.6	11.9	9.0	21.7	24.4	4.1	6.6	6.6	12.2	12.4	5342

**Anti-AT1R**: anti-angiotensin 1 receptor autoantibody; **anti-ETAR**: anti-endothelin receptor autoantibody; **anti-α1 adrenerg**: anti-α1 adrenergic receptor autoantibody; **anti-α2 adrenerg**: anti-α2 adrenergic receptor autoantibody; **anti-β1 adrenerg**: anti-β1 adrenergic receptor autoantibody; **anti-β2 adrenerg**: anti-β2 adrenergic receptor autoantibody; **anti-MC R1**: anti-muscarinic cholinergic receptor-1 autoantibody; **anti-MC R2**: anti-muscarinic cholinergic receptor-2-autoantibody; **anti-MC R3**: anti-muscarinic cholinergic receptor-3 autoantibody; **anti-MC R4**: anti-muscarinic cholinergic receptor-4 autoantibody; **anti-MC R5**: anti-muscarinic cholinergic receptor-5 autoantibody; **TPO**: anti-thyroid peroxidase autoantibody.

## Data Availability

Autoantibody data in children with MIS-C are currently published: Buchhorn, R.; Meyer, C.; Heidecke, H.; Willaschek, C. Functional autoantibodies against G-protein coupled receptors in children with complications after SARS-CoV-2 infections. *Isr. Med. Assoc. J.*
**2021**, (accepted, in press).

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
