# Peer review of "Autoantibody Release in Children after Corona Virus mRNA Vaccination: A Risk Factor of Multisystem Inflammatory Syndrome?"

_vaccines, 2021, doi:10.3390/vaccines9111353_

Round 1
Reviewer 1 Report
The reviewer thanks the authors for their revised version. However, they did not totaly answer to my questions/
Because of the great variations in laboratory methodologies to detect auto-antibodies, it would be important to precize the kit used is industrial or an in-house assay.
The absence of sample before vaccination for case 1 is a severe limitation to the conclusions of the authors who claim that the antibodies are related to the vaccination. This should be clearly indicated in the discussion.
When we look at the profiles of the antibodies post vaccination in comparison to the antibodies post-COVID disease, in both cases anti MC R3 and R4 are anly minimally increased while they are clearly increase in post-COVID disease. This should be mentionned and discussed (if the authors have an explanation)
Sincerely yours
Author Response
Dear collaegue, with respect to laboratory methodologies I include a methodological paragraph in the introduction (green) and discuss the limitations from missing baseline values in case 1:
Moreover, we do not know the baseline values before vaccination in this boy who had pre-existing illness.
We declare the difference in MC R3+R4 autoantibodies but we can not draw general conclusions of cause the low number of patients:
However, anti α 1 – and anti muscarinergic receptor 3+4 autoantobodies are more elevated in 4 children after SARS-COV-2 infection compared to this case. The number of cases is to low in order to be able to draw general conclusions from this.
Reviewer 2 Report
The revised version of the manuscript is, in my opinion better than the previous. However, there are still some points to be clarified 1) Describing CASE 1 the authors affirm "A genetic autoinflammatory syndrome was excluded by the university hospital." Could they explain how colleagues ruled out a genetic problem? 2)I couldn't find the manuscript that the table 162.b2 refers to. Could the authors explain on how many samples of healthy subjects the normal values ​​were established?If the authors had used a commercial kit (indicate its name and manufacturer) they should emphasize that the normal values ​​are indicated by the manufacturer and in any case distinguish the values ​​defined "at risk" from those identified as "positive" i.e. Celltrend EIA for Quantitative Determination of anti-beta2-adrenergic Receptor Antibodies , cut off (Löbel et al, Brain Behav Immun 2015):
positive (95% percentile): 14.0 Units/ml at risk (90% percentile): 8.0 – 14.0 Units/ml
Author Response
Thank you for the review.
With respect to the genetic disorder we include: ........by the university hospital (tumor necrosis factor receptor associated periodic syndrome, familial mediterranean fever)
The table was incompletely includes by the journal. We include the number of MIS-C cases (N=4)
We include a methodological paragraph in the introduction (green labeled) and the at risk values so far known in the table.
Round 2
Reviewer 1 Report
No comment
Author Response
Dear reviewer,
Thank you again to give me the opportunity to revise our manuscript. With respect to your comments. I include the following sentences in the paper:
- A few case reports have been published on MIS after SARS-CoV-2 vaccination. However, these are very cautious and refer to symptoms developed in a relatively short time. Ten weeks duration to see elevated autoantibodies in your study during which many events could have influenced patients. symptoms. Please comment and revise your manuscript.
.... the diagnosis was made late and remain unsure. However, the guidelines define a time frame for onset of MIS-C lower than 12 weeks post-infection/vaccination, although MIS-C cases predominantly present 4-6 weeks following COVID-19. Moreover, a specialized clinic didn’t find any other explanation for the inflammatory syndrome of this boy, which is not doubted. It clearly corresponds to our duty of care, that we reported the unwanted side effect of BNT162b2 to the responsible drug authorities.
- Sample size is too small as well as lacks proper methodology (levels of autoantibodies before vaccination) to arrive at a conclusion. You may discuss it in the manuscript.
We would like to emphasize again that sample size is too small as well as lacks proper methodology (levels of autoantibodies before vaccination in case 1) to arrive a final conclusion. However, our data has encouraged us to prospectively investigate the formation of autoantibodies against G-protein coupled receptors after various SARS-CoV-2 vaccinations.
- Revise your claims in the discussion part considering the fact that many countries are in the process of approving the vaccination for children. You may focus on the importance of management and therapy in rare cases if MIS happens after vaccination.
Many countries are in the process of approving the vaccination for all children because the health policy goal of achieving herd immunity seems only be achieved by immunizing the population as completely as possible. Although the complications of vaccination are apparently rare, we may focus on the importance of management and therapy of this rare cases if we want to maintain public acceptance of the SARS-CoV-2 vaccination.
Some minor changes are underlined in grey colour.
Yours sincerely
Prof. Reiner Buchhorn

This manuscript is a resubmission of an earlier submission. The following is a list of the peer review reports and author responses from that submission.
Round 1
Reviewer 1 Report
The authors reported two cases referable to multisystem inflammatory syndrome after SARS-COV-2 vaccination.
“The first case refers to a 18 year old boy who suffers from hypoxic ischemic encephalopathy after a complicated birth and receives pharmacotherapy due to his epilepsy and tetraspastic who developed inflammatory syndrome 10 weeks after vaccination.”
As the authors point out “There is a high burden of inflammatory disease in this family: The brother of the 115patient had pericardial effusion of unknown origin six years ago, treated with colchicine 116for 6 months. He received a corona virus vaccination (BNT162b2) without significant side 117effects and without a relapse of his pericardial effusion. The mother of the boy recently suffers from inflammation with pericardial effusions without known COVID-19 or vaccination.”
The same antibodies tests should also be performed on the brother, but above all on the mother to rule out that there is a genetic component in the development of the autoantibodies.
Reviewer 2 Report
In this paper, the authors describe 2 cases of auto-antibodies against G-protein coupled receptors in 2 children, after SARS-CoV-2 vaccination
Major comments:
- In the introduction section, the authors do not explain why they focused their study on this type of antibodies, which are not very frequently explored in routine biology
- Did they quantify more classical auto-antibodies such as anticardiolipin and anti-Beta2 glycoprotein which were frequently studied in different reports on SARS-CoV-2 infection ?
- It is well known that in some cases, auto-antibodies can be observed in healthy subjects, without evident disease; Do the authors have an idea of the frequency of such cases in healthy population ?
- If the increase in auto-antibodies titer is convincing for case 2, since the authors have a titer of antibodies before the vaccination, the absence of basal levels for the case 1 does not allow to conclude that the appearance of these auto-antibodies are related to the vaccination. This a severe limitation which shoud be indicated and discussed.
- The discussion is too long and not always related to the data presented. For example, page 7, the paragraph from line 227 to 235 is not related to the data, or should be indicated in the introduction section, as suggested before
- in fact, vaccination stimulates the immune system, therefore it not really surprising that the titer of pre-existing auto-antibodies increase. Did the authors studied auto-antibodies levels variations adults, for example ?
- search for auto-antibodies is highly dependent of the assay used. The assay used should be detailed (home or commmercail assay ?)
Minor comments:
- legends from figures and tables should be improved
- reference ranges for biological assays should be indicated
- name of medicines do not need a capital character